# A Comparative Study of Resistant Dextrins and Resistant Maltodextrins from Different Tuber Crop Starches

**DOI:** 10.3390/polym15234545

**Published:** 2023-11-27

**Authors:** Xinyang Chen, Yinchen Hou, Zhen Wang, Aimei Liao, Long Pan, Mingyi Zhang, Yingchun Xue, Jingjing Wang, Yingying Liu, Jihong Huang

**Affiliations:** 1Henan Key Laboratory of Wheat Bioprocessing and Nutritional Function, School of Biological Engineering, Henan University of Technology, Zhengzhou 450001, China; 2021930593@stu.haut.edu.cn (X.C.); diamond004@163.com (Y.X.);; 2School of Food and Bioengineering, Henan University of Animal Husbandry and Economy, Zhengzhou 450046, China; 3State Key Laboratory of Crop Stress Adaptation and Improvement, College of Agriculture, Henan University, Kaifeng 475004, China; 4Food Laboratory of Zhongyuan, Luohe 462300, China; 5School of Food and Pharmacy, Xuchang University, Xuchang 461000, China

**Keywords:** thermal-acid treatment, resistant dextrins, resistant maltodextrins, structure characterization, in vitro digestibility

## Abstract

The anti-digestibility of resistant dextrin (RD) and resistant maltodextrin (RMD) is usually significantly affected by processing techniques, reaction conditions, and starch sources. The objective of this investigation is to elucidate the similarities and differences in the anti-digestive properties of RD and RMD prepared from three different tuber crop starches, namely, potato, cassava, and sweet potato, and to reveal the associated mechanisms. The results show that all RMDs have a microstructure characterized by irregular fragmentation and porous surfaces, no longer maintaining the original crystalline structure of starches. Conversely, RDs preserve the structural morphology of starches, featuring rough surfaces and similar crystalline structures. RDs exhibite hydrolysis rates of approximately 40%, whereas RMDs displaye rates lower than 8%. This disparity can be attributed to the reduction of α-1,4 and α-1,6 bonds and the development of a highly branched spatial structure in RMDs. The indigestible components of the three types of RDs range from 34% to 37%, whereas RMDs vary from 80% to 85%, with potato resistant maltodextrin displaying the highest content (84.96%, *p* < 0.05). In conclusion, there are significant differences in the processing performances between different tuber crop starches. For the preparation of RMDs, potato starch seems to be superior to sweet potato and cassava starches. These attributes lay the foundation for considering RDs and RMDs as suitable components for liquid beverages, solid dietary fiber supplements, and low glycemic index (low-GI) products.

## 1. Introduction

Dietary fiber has received considerable attention due to its positive effects on the prevention and/or treatment of type 2 diabetes, cardiovascular disease, intestinal disorders, and related complications [1,2]. Dietary fibers come from a wide range of sources, encompassing almost all aggregates formed with glucose as the basic unit; for example, resistant dextrin (RD) and resistant maltodextrin (RMD) are defined as dietary fibers by the U.S. Food and Drug Administration [3]. The development of dietary fibers from starch is considered to be sustainable and has therefore aroused the interest of cereal scientists in the process of the chemical modification of starch for the development of new dietary fiber formulations.

Generally, products of starch dextrinization conducted under specific conditions, including dry heat, thermal-acid, and microwave methods, may be resistant to enzyme digestion [4,5]. RD, often referred to as pyrodextrin, is a starch-derived product that exhibits partial resistance to enzymatic hydrolysis in the human gastrointestinal tract [6]. In addition to the α-1,4- and α-1,6-glycosidic bonds found in starch molecules, this highly branched carbohydrate polymer contains relatively short glucose oligomer chains/polymer chains that are linked to many types of indigestible glycosidic bonds [7,8]. RDs derived from various starch sources are usually obtained under harsh dextrinization conditions using high acid concentrations (≥0.1% HCl or H_2_SO_4_), heating temperatures (≥130 °C), and extended heating times (≥60 min) [9,10]. In the case of chemically modified starch, a resistance to enzymatic digestion is caused by spatial site resistance formed at the site of enzymatic action. Specifically, 1,2- and 1,3-glycosidic bonds (and perhaps new 1,6-glycosidic bonds) are produced during starch dextrinization at the expense of converting the 1,4- and 1,6-glycosidic bonds characteristic of starch [11]. Namely, RD production is the result of a complex process that occurs under the influence of temperature and acid catalysts on starch, including depolymerization, transglucosylation, and repolymerization. The formation of new bonds makes dextrins less sensitive to digestive enzyme activity by reducing the number of targets for potential attack [12]. Thus, starch molecules with α-1,2- and α-1,3-bonds or highly aggregated bonds exhibit functional properties (anti-digestive properties) similar to those of dietary fiber and prebiotics [13]. Many physiological benefits of this starch-derived dietary fiber have also been demonstrated; these include the ability to lower the glycemic effect of foods, lower plasma triglyceride levels, increase absorption, and the retention of minerals and their prebiotic activity [14]. The further hydrolysis of RD with α-amylase is able to produce α-limit dextrins with glucose equivalents of the hydrolysis product of less than 20; the resulting products are known as resistant maltodextrins (RMDs) [15,16]. RDs and RMDs are used in a wide range of functional food and beverage products because both compounds have excellent water solubility and thermal stability characteristics, provide low viscosity for a variety of food matrices, and can tolerate high temperatures and low pH levels. 

Tuber crops are widely cultivated globally, with high starch content and relatively simple extraction methods. Therefore, tuber crops can provide sufficient raw materials for the large-scale production of RD and RMD. Potato (*Solanum tuberosum* L.), cassava (*Manihot esculenta* L.), and sweet potato (*Ipomoea batatas* L.) are the most common tuber crops grown globally, and previous studies have fully reported the molecular structure behavior and changes in product properties during the preparation of RDs and/or RMDs from potato, cassava, and sweet potato starches [5,17,18,19,20]. However, to the best of our knowledge, there is no comparative cross-sectional study of the three potato starches under the same RD/RMD preparation conditions, nor is there a lack of in-depth reports of the longitudinal comparisons (production of RDs from starch feedstock and further production of RMDs) of the three different tuber starches. Therefore, we employ identical processing technologies and conditions to treat three distinct starches for the fabrication of resistant dextrins (RDs) and resistant maltodextrins (RMDs). This approach allows us to investigate the influence of different starches on the properties of RDs and RMDs. The data presented in this study not only demonstrates the relationship between the structural characteristics of these compounds, but also provides insights into their in vitro digestibility. Furthermore, we conduct analyses on their solubilities, pasting properties, and thermal properties to achieve a more comprehensive understanding.

## 2. Materials and Methods

### 2.1. Materials

Normal sweet potato (*Ipomoea batatas* L.), cassava (*Manihot esculenta* L.), and potato (*Solanum tuberosum* L.) starches were obtained from Liangrun Whole Grain Food Co., Ltd. (Xinxiang, China). Glucose content kits (GOPOD) were purchased from Jiancheng Bioengineering Institute (Nanjing, China). Thermostable α-amylase (40,000 U/g) and amyloglucosidase (100,000 U/g) were purchased from Macklin Biochemical Technology Co., Ltd. (Shanghai, China). α-Amylase (from pig pancreas, ≥5 μ/mg solid) was purchased from Yuanye Bio-Technology Co., Ltd. (Shanghai, China). All other chemicals were of an analytical grade.

### 2.2. Preparation of Samples

#### 2.2.1. Preparation of RDs

RDs were prepared by the method of Bai et al. [21] with some modifications. Briefly, the starch (10 g) was suspended in 15 mL of 0.06 mol·L^−1^ HCl. After 30 min, the mixture was centrifuged at 4000 rpm for 10 min. Then, the precipitate was dried at 40 °C for 24 h and pyrolyzed at 170 °C for 1.5 h to produce RDs. Normal sweet potato, cassava, and potato starches were used in the same process conditions to prepare the RDs.

#### 2.2.2. Preparation of RMDs

The RD was dispersed in water (1:3, *w*/*v*). The pH was adjusted to 6.0 by using 0.1 M of NaOH, thermostable α-amylase (1%, *m*/*m*) was added and stirred (95 °C, 2 h), and then the pH was adjusted to 4.5 using 0.1 M of HCl, and amyloglucosidase (0.5%, *v*/*m*) was added and stirred (60 °C, 1 h). The final product was concentrated and the supernatant was added to 4 times the volume of liquid 95% ethanol. The sample was left to rest for 4 h and the precipitate was dried to constant weight to obtain the RMD [19,22]. The same process conditions were used for the preparation of the three RMDs.

### 2.3. Structure Characterization

#### 2.3.1. Scanning Electron Microscopy (SEM)

The morphological characteristics of the starches, RDs, and RMDs were visualized by SEM (Quanta250FEG, FEI, Brno, The Czech Republic). The sample was dispersed on the sample holder and sprayed with gold by an ion sputtering apparatus, and the magnification was adjusted to 2000× or 3000× to observe the morphology [23].

#### 2.3.2. Fourier Transform Infrared Spectra (FTIR)

An infrared spectrogram was obtained using FTIR (Nicolet iS20, Thermo Fisher Scientific, Wilmington, DE, USA). The sample was accurately weighed and mixed with potassium bromide at a ratio of 1:100 (*m*/*m*), and then the mixture was pressed into flakes. Finally, the samples were scanned at 4000 to 400 cm^−1^, with 32 scans and a resolution of 4 cm^−1^ [24].

#### 2.3.3. X-ray Diffraction (XRD)

XRD was used to determine the crystalline structure of the starches, RDs, and RMDs [25]. The diffraction was performed utilizing a wide-angle diffractometer, scanning within the range of 5° to 60° (2θ) (MiniFlex600, Rigaku, Tokyo, Japan). 

#### 2.3.4. Nuclear Magnetic Resonance Spectra (NMR)

The types of glycosidic bonds in the RDs and RMDs were determined by the method of Bai et al. [21]. The sample (0.1 g) was exchanged twice with D_2_O (1 mL), and the sample, after deuterium exchange was dissolved in D_2_O (99.9%) with a final concentration of 10% (*w*/*v*). The analysis was performed by a NMR system (AVANCE III HD 600 MHz, Bruker, Karlsruhe, Switzerland) at 25 °C. Subsequently, ^1^H spectra were collected in 32 separate scans with a scan width of 16 ppm and a delay time of 1 s. Tetramethylsilane at 0 ppm was used as an internal reference.

### 2.4. Physicochemical Properties

#### 2.4.1. Pasting Property

The sample was dispersed in water (3:25, *m*/*v*) and then apparent viscosity measurements were evaluated using a Rapid Visco Analyzer (RVA 4800, Perten, Stockholm, Sweden). The specific reaction procedure was referenced from Li et al. [10]. 

#### 2.4.2. Thermal Properties

The thermal properties of the RDs and RMDs were meticulously evaluated using a differential scanning calorimeter (DSCQ20, TA, New Castle, PA, USA) and a thermogravimetric analyzer (TGQ600, TA, New Castle, PA, USA). The sample (2 mg) was mixed with distilled water (6 μL) in a DSC aluminum pan. Prior to the analysis, the pan was sealed and allowed to equilibrate at room temperature for 12 h. Subsequently, the analysis involved heating the sample from 30 to 100 °C at a rate of 10 °C/min, with an empty pan serving as a reference. The resulting heat flow curve was then recorded [26]. A thermogravimetric analysis was performed according to the following procedure: the heating rate was set at 10 °C/min from 30 °C to 400 °C, with a carrier gas nitrogen flow rate of 20 mL/min, and the weight loss curve was recorded during the heating process [22].

#### 2.4.3. Solubility 

The sample (0.4 g) was suspended in deionized water (20 mL). The solution was heated at 25 °C for 30 min and centrifuged at 11,600× *g* for 10 min. Then, the supernatant (10 mL) was heated at 105 °C for 4 h [10]. The solubility of the sample was calculated using the following equation: (1)$$\mathrm{Solubility}(\%)=\frac{20\times \mathrm{Soluble}\;\mathrm{sample}\;\mathrm{weight}}{10\times \mathrm{Sample}\;\mathrm{weight}}\times 100\%$$

#### 2.4.4. Indigestible Ingredient Content

The determination of the indigestible ingredient content was conducted following Matsuda’s method with some modifications [27]. Initially, a 50 mL phosphate-buffered solution of 0.05 mol^−1^·L and pH 6.0 were used to suspend the sample of 0.25 g (constant weight). Subsequently, a thermostable α-amylase (60 U/mL) sample of 1.0 mL was added and stirred well (95 °C, 30 min), the resulting mixture was cooled to room temperature, and the pH was adjusted to 4.5 using 0.1 M of HCl. Then, an amyloglucosidase (100 U/mL) sample of 1.0 mL was added and stirred (60 °C, 30 min). Finally, the mixture was inactivated at 100 °C for 10 min and adjusted to a final volume of 100 mL. The glucose content of the sample was determined using the DNS colorimetric method.
(2)$$\mathrm{Content}\;\mathrm{of}\;\mathrm{indigestible}\;\mathrm{component}\;\left(\%\right)=100\;-\mathrm{Amount}\;\mathrm{of}\;\mathrm{glucose}\;\mathrm{formed}\;\left(\%\right)\times 0.9$$

### 2.5. In Vitro Digestibility

The in vitro digestibility of the sample was determined using Englyst’s model with slight modifications. The sample (0.20 g) was added into 20 mL of 0.5M sodium acetate buffer (pH = 5.2), and the suspension was gelatinized in boiling water for 30 min. After cooling at room temperature, 5 mL of mixed enzyme liquids (containing 290 U/mL of pig pancreatic α-amylase and 20 U/mL of amyloglucosidase) were added and placed in a constant-temperature water-bath shaker at 37 °C, and the mixture was fully shaken at 190 rpm. The glucose contents were determined by the GOPOD kit at 20, 60, 90, 120, 150, and 180 min, the hydrolysis rates were calculated, and the starch hydrolysis data were fitted to a first-order equation [28].
(3)$$\mathrm{Hydrolysis}\;\mathrm{rate}\;(\%)=\frac{\mathrm{Reducing}\;\mathrm{sugar}\;\mathrm{content}}{\mathrm{Dry}\;\mathrm{matter}}\times 100\%$$
(4)$${Ct}_{}=C_{\infty }\left(1\;-\;e^{-\mathrm{kt}}\right)$$

C_t_ is the concentration of the product or reactant at time t, C_∞_ is the corresponding concentration at the end point, and k is a pseudo first-order rate constant. 

### 2.6. Statistical Analysis 

All experiments were conducted in triplicate and the results are expressed as the mean ± standard deviation. OriginPro 2023b (OriginLab, Northampton, MA, USA) was used for the data visualization. Statistical analyses were performed using SPSS 25 (IBM, Armonk, NY, USA) with ANOVA, followed by Duncan’s *post hoc* test (*p* < 0.05). 

## 3. Results and Discussion

### 3.1. Morphological Analysis

The morphological characteristics of the samples were shown in Figure 1. The surface structure of the sweet potato starch was smooth, spherical, or nearly spherical. The surface structure of the potato starch was complete and smooth, forming an irregular oval. Cassava starch had an egg shape with a smooth surface. These morphological characteristics were consistent with the other reported results [29,30,31]. Meanwhile, all RDs were prepared through a thermal-acid reaction, and the typical granular appearance of native starch was not destroyed, keeping consistency with the report [5,9,10]. Specifically, all RD samples showed no macroscopic changes compared to the starch group, but the slight hydrolysis on the surface of the starch particles needed to be noted. This may indicate that the source of the anti-digestion ability of RDs is not significantly related to changes in the starch particle morphology. However, the surface of the RMD samples exhibited numerous fragmented structures with noticeable holes and fissures. After undergoing enzymatic purifications in two steps, the structures of RMDs transformed into irregular forms rich in cavities. Similar structural alterations were consistent with the findings reported by Chen et al. [26]. Concurrently, starch granules were degraded, leading to the repolymerization of small molecules and glycosylation reactions. These transformations resulted in structural changes that facilitated the penetration of water molecules, significantly enhancing water solubility [22]. Additionally, the changes in the molecular structure are detailed in the following section.

### 3.2. FTIR Spectra Analysis

The FTIR spectra of RDs, RMDs, and natural starches revealed differences in their molecular structures (Figure 2), which could provide some explanations for the hydrolysis rates of starches, RDs, and RMDs. By comparing the FTIR spectra of starches, RDs, and RMDs, it was found that the bands near 1652 cm^−1^ were different, and the band should be attributed to the affinity of water molecules within the starch molecules [32]. Meanwhile, the peaks and valleys of the spectrum significantly indicated an increase in the hydrophilicity of RMDs at 3358 cm^−1^. Based on the morphological analysis, these changes are easy to understand. The peak intensities of all RDs and RMDs at 928 cm^−1^ differed from their respective counterparts in natural starches. The dextrinization process led to a decrease in the number of α-1,4 glycosidic bonds, which was the primary reason for the formation of anti-enzyme digestion properties in RDs and RMDs. This decrease can be explained by the reduction in the molecular weights of RDs and RMDs, as well as by the transglycosylation process, where α-1,4-glycosidic bonds were replaced by other bonds. The transglycosylation process may be one of the main reasons for the decrease in the digestibility of RDs and RMDs [19,21]. Under thermal-acid conditions, starch molecules underwent the processes of de-chaining and re-polymerization, leading to the breakdown of α-1,4-glycosidic bonds within the starch molecules. This interaction initiated the cross-linked polymerization of shorter chains within the starch molecules. This interaction may have triggered the cross-linking polymerization of short chains within the starch molecule, resulting in the formation of new glycosidic bonds within the molecule, such as α-1,2, β-1,2, etc.; these newly formed bonds exhibited anti-digestive properties [24]. The analysis revealed the decrease in the specified ratios (1047 cm^−1^/1022 cm^−1^ and 1022 cm^−1^/995 cm^−1^) when comparing RD to starch, suggesting changes in the molecular composition or structure. The observed changes included the disruption of the short-range ordered structure caused by the thermal-acid reaction. Furthermore, the ratio of the amorphous polymer structure to ordered polymer structure in the starch was decreased after this reaction.

### 3.3. XRD Analysis

By analyzing the XRD patterns of the starches, RDs, and RMDs in Figure 3, it can be observed that sweet potato starch displayed diffraction peaks at 15.2°, 17.1°, and 23.1° (2θ) with sharp peaks, indicating a typical A-type crystal structure. Similarly, potato starch exhibited diffraction peaks at 5.6°, 14.6°, 17°, and 22.4° with sharp features, signifying a typical B-type crystal structure. Lastly, cassava starch demonstrateed diffraction peaks at 15.1°, 17.1°, 17.8°, and 23.0°, corresponding to a typical A-type crystal structure. Similar results have previously been reported [33,34,35]. RDs also exhibited the aforementioned absorption peaks, albeit with reduced sharpness. Following the enzymatic treatments, the crystal structures of RMDs significantly changed, becoming distinctly different from those of the starches. All RMDs showed broad diffraction peaks, with the sharp features nearly disappearing, indicating a transition from crystalline to amorphous structures. The repolymerization of small molecules and intermolecular glycosyl transfer reactions were promoted by thermal-acid reactions. As a result, regular intermolecular arrangements were disrupted, weakening the intermolecular forces and hydrogen bonds, leading to a reduction in the crystallization ability. However, if the starches were only subjected to thermal-acid treatment, their crystalline structures would not be completely destroyed, staying consistent with the report by Bai et al. [35]. Subsequently, the α-1,4 and α-1,6 glycosidic bonds were further hydrolyzed by enzymes. The crystallization peaks disappeared, replaced by a diffuse pattern, indicating the disruption of the ordered crystal structure and its transformation into an amorphous state.

### 3.4. NMR Spectra Analysis

Nuclear magnetic resonance hydrogen spectroscopy was used to identify the types of glycosidic bonds in both the RDs and RMDs. As shown in Figure 4, the signals between 4.4 and 5.5 ppm originate from anomeric protons and were well separated. For the ^1^H NMR spectra of the RDs and RMDs, not only α-1,4 and α-1,6 glycosidic bonds were found, but also new bonds or linkages were observed, formed as a result of starch hydrolysis, denaturation, glycosylation, and repolymerization processes. They included 1,6-anhydrous-β-D-glucopyranose, α-1,2, β-1,2, β-1,4, and β-1,6 linkages, each appearing at distinct chemical shift positions. These findings are consistent with the previous research on RDs and RMDs, although there may be slight variations due to the differences in the preparation methods employed [21]. In the context of specific processing conditions, different thermal-acid temperatures and times, as identified in a previous study by Han et al. [36], strongly influenced the structural characteristics of the prepared pyrodextrin. With an increase in the heating time, there was a notable increase in β-1,6-anhydro, α-1,2, α-1,6, β-1,2, and β-1,6 bonds, while the level of the α-1,4 linkage decreased. Additionally, previous studies have revealed that heightened microwave power intensity and heating time were conducive to the promotion of molecular branching [5].

The digestibility and potential applications of RDs and RMDs were greatly influenced by their molecular structures. The indigestible components of RDs and RMDs were generated through dextrinization, and notably, RMDs underwent further purification via α-amylase treatment, leading to a decreased proportion of α-1,4 glycosidic bonds. Despite the presence of numerous α-1,4 linkages in the RDs and RMDs, their resistance to enzymatic digestion remained notably high, likely due to the highly branching structure of the RDs and RMDs [19].

### 3.5. Pasting Properties

The pasting property curves of starches, RDs, and RMDs were shown in Figure 5. The determination of pasting properties using an RVA was a remarkably important tool for the development of low-viscosity products. All three starches exhibited typical starch pasting curves. In contrast, both RDs and RMDs remained stable and displayed low viscosity levels. It was observed that none of the RD and RMD samples exhibited significant viscosity developments, and their viscosities were almost negligible compared to the natural starches. This observation highlighted the substantial molecular degradation resulting from the thermal-acid treatment [37]. These findings suggest that the dextrinization process has a significant influence on the swelling capacity of starches and the development of viscosity, and this effect can be attributed to the reduction in the starch molecular size and crystallinity during the dextrinization process [38]. This result was consistent with the structural changes observed through the SEM and XRD. Le’s study reported that, when the water solubility of pyrodextrin exceeded 90%, it maintained its Newtonian fluid behavior, even at a material–liquid ratio of 40% (*m*/*v*) [24].

### 3.6. Thermal Properties Analysis

There were no phase transition peaks observed in the prepared RDs and RMDs (Figure 6A). This phenomenon may be attributed to the potential alteration or loss of the original crystal structure of amylopectin following thermal-acid reactions. The outcomes obtained from the XRD, SEM, and RVA analyses align with this observation. The thermal stability levels of the RDs and RMDs were investigated by programmed heating in the range of 30 to 400 °C. Derivative thermogravimetry (DTG) was used to assess the rate of mass loss during the heating of the samples. As shown in Figure 6B,C, all RDs and RMDs demonstrated relative stability rates below 200 °C. However, a notable feature on the DTG curve was observed with a sharp peak occurring at approximately 300°, which was the degradation temperature of the RDs and RMDs. The potential industrial applications of RDs and RMDs were closely related to their thermal properties, relying on their good thermal stability to maintain the nutritional quality and physicochemical properties of foods during various thermal processing methods, including heat processing and baking [22]. 

### 3.7. Solubility and Indigestible Ingredient Content Analyses

The solubilities and indigestible ingredient contents of the RDs and RMDs were shown in Table 1. The solubilities of the three starches were only approximately 1.5% at room temperature. In contrast, the solubilities of RMDs were higher than those of RDs and starches at room temperature. Specifically, the solubility values of sweet potato, potato, and cassava increased by 32.57%, 33.66%, and 38.55%, respectively (*p* < 0.05), when the samples were converted from RD to RMD. The increased solubilities of RMDs indicated that the starches were broken down into smaller compounds during the pyrolytic conversion, while the α,1–4 and α,1–6 bonds in the RDs were hydrolyzed by α-amylase and amyloglucosidase. These enzymatic actions resulted in a higher proportion of oligosaccharides, a lower degree of polymerization, and consequently, higher water solubilities [39]. On the other hand, the information provided by the FTIR (Figure 2) and XRD (Figure 3) suggested that the decrease in the degree of ordering and crystal structure in the starch molecules might have been an important reason for the increased solubility of the RMDs. The mechanistic relationship between the decrease in the crystal structure of starch and the increase in solubility can be understood as an inverse relationship. The crystal structure of starch is an ordered arrangementof multiple glucose units connected by glycosidic bonds. When the crystal structure of starch is reduced, it means that the ordered arrangement of starch molecules is disrupted, and the hydrogen bonds and other interactions within the molecules are weakened, resulting in the tight structure of starch molecules becoming loose. This loosening of the structure results in an increased interaction between the starch and water molecules. Water molecules are able to penetrate more easily into a starch molecule and form hydrogen bonds with the hydrophilic groups on its surface, thus gradually dissolving the starch molecule in water. Thus, as the crystal structure of starch decreases, the solubilities of RDs and RMDs in water increase. Because of their excellent solubilities and indigestible ingredient contents, RMDs have been reported for use in producing prepared soups, low-calorie beverages, and beers with a high dietary-fiber content [40,41].

### 3.8. In Vitro Digestibility

The hydrolysis rates of starches, RDs, and RMDs were shown in Figure 7 and Table 2. Within 20~180 min, the glucose contents in the digestion products of a1, b1, and c1 were significantly increased, and the hydrolysis rates of the corresponding RDs stabilized at approximately 40% around the 60 min mark, whereas the corresponding RMDs exhibited a slow increase, with all of their hydrolysis rates remaining below 8% throughout the digestion period. All the samples exhibited excellent fitting with R^2^ values exceeding 0.9, except for the potato resistant dextrin. Additionally, our observations revealed that, among the three starches, potato starch displayed the slowest digestion rate, potentially attributed to its crystalline structure. Previous research has indicated that starches with a B-type crystalline structure tend to have a slower hydrolysis rate than to those with an A-type structure [42]. The rapid hydrolysis rates of RDs may be attributed to the thermal-acid reaction, starch de-chaining, and the subsequent release of enzyme binding sites, leading to a high initial hydrolysis rate. However, it was noteworthy that these hydrolysis rates stabilized shortly thereafter. Simultaneously, we noted that the hydrolysis rates of RMDs were notably low. The exceptionally low hydrolysis rates of RMDs were in line with the desirable characteristic of maintaining postprandial blood glucose levels, aligning with the recognized functional properties associated with dietary fibers [43].

All RDs and RMDs contained substantial quantities of indigestible components, primarily attributed to the dextrinization of starches during thermal-acid treatment. This process resulted in the creation of spatial resistance sites at the enzyme’s active site and the formation of glycosides with β-1,2 and β-1,4 bonds derived from α-1,4 bonds, among other structural modifications. These alterations rendered the RDs and RMDs less susceptible to the activity of digestive enzymes, contributing to their reduced digestibility [18]. The quantities of indigestible components displayed an inverse relationship with the levels of α-1,4 bonds, given that the primary digestive enzyme in humans, α-amylase, primarily targeted α-1,4 glycosidic bonds. The production of RMDs was derived from the RDs, and the RDs were purified by the two-step enzymatic reactions, resulting in a reduced presence of α-1,4 glycosidic bonds in the RMDs [8]. Therefore, it was anticipated that the postprandial glycemic responses for all RMDs would be reduced due to their low glucose contents in digestive products, limited hydrolysis, and a lower proportion of glycosidic bonds (α-1,4 and α-1,6 bonds).

## 4. Conclusions

The structures of RDs and RMDs were significantly different from those of natural starches, exhibiting disordered spatial arrangements, destroyed crystal structures, and a loss of the ability to form pastes. While the microstructure of the RDs still retained starch-like characteristics, the RMDs developed surface irregularities and porous structures. Meanwhile, ^1^H NMR and FTIR spectra revealed a reduction in α-1,4 glycosidic bonds during the reaction, accompanied by the formation of new glycosyl links, including α-1,2, β-1,2, β-1,4, and β-1,6 bonds. The indigestible components of the three types of RDs ranged from 34% to 37%, whereas RMDs varied from 80% to 85%, with potato resistant maltodextrin displaying the highest content (84.96%). All RDs and RMDs exhibited favorable thermal stability (<200 °C) and high solubility (>60%). These properties provided a theoretical foundation for the application of RD and RMD in various food products, including, but not limited to, cookies, low-glycemic-index reconstituted rice products, and dietary-fiber-enriched beverages. In addition, the forthcoming research will center on evaluating the effects of incorporating RDs and RMDs into starch-based foods, including their effects on pasting, aging, rheological properties, and digestive behavior. We will also investigate the potential for directed dextrinization to enhance the levels of indigestible glycosidic bonds. This research will play a pivotal role in advancing sweet potato (*Ipomoea batatas* L.), cassava (*Manihot esculenta* L.), and potato (*Solanum tuberosum* L.) industries, contributing to their high-quality growth.

## Figures and Tables

**Figure 1 polymers-15-04545-f001:**
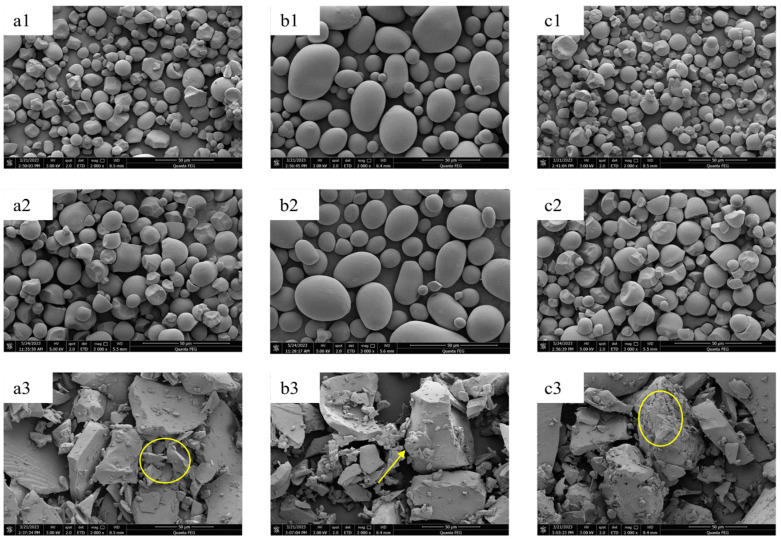
Scanning electron microscopy images of starches, resistant dextrins, and resistant maltodextrins: (**a1**) sweet potato starch; (**a2**) sweet potato resistant dextrin; (**a3**) sweet potato resistant maltodextrin; (**b1**) potato starch; (**b2**) potato resistant dextrin; (**b3**) potato resistant maltodextrin; (**c1**) cassava starch; (**c2**) cassava resistant dextrin; (**c3**) cassava resistant maltodextrin. Surface holes on RMDs, delineated by arrow and circles.

**Figure 2 polymers-15-04545-f002:**
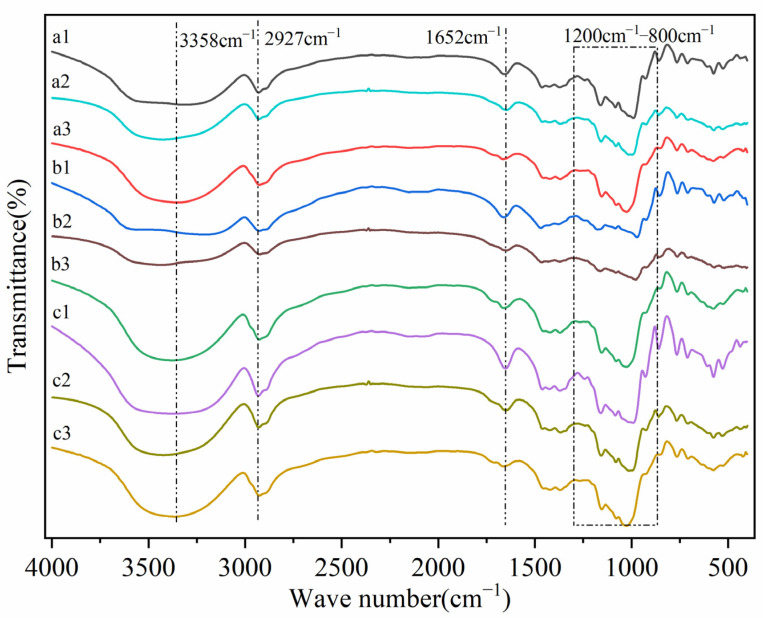
FTIR spectra of starches, resistant dextrins, and resistant maltodextrins: a1: sweet potato starch; a2: sweet potato resistant dextrin; a3: sweet potato resistant maltodextrin; b1: potato starch; b2: potato resistant dextrin; b3: potato resistant maltodextrin; c1: cassava starch; c2: cassava resistant dextrin; c3: cassava resistant maltodextrin.

**Figure 3 polymers-15-04545-f003:**
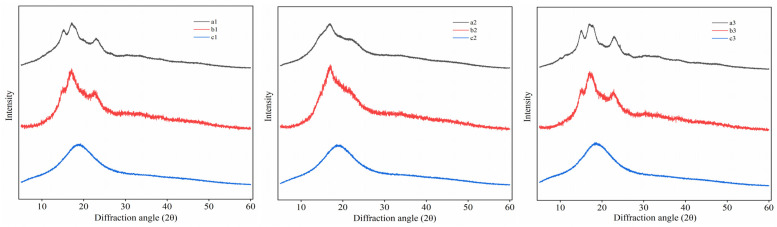
XRD patterns of starches, resistant dextrins, and resistant maltodextrins: a1: sweet potato starch; a2: sweet potato resistant dextrin; a3: sweet potato resistant maltodextrin; b1: potato starch; b2: potato resistant dextrin; b3: potato resistant maltodextrin; c1: cassava starch; c2: cassava resistant dextrin; c3: cassava resistant maltodextrin.

**Figure 4 polymers-15-04545-f004:**
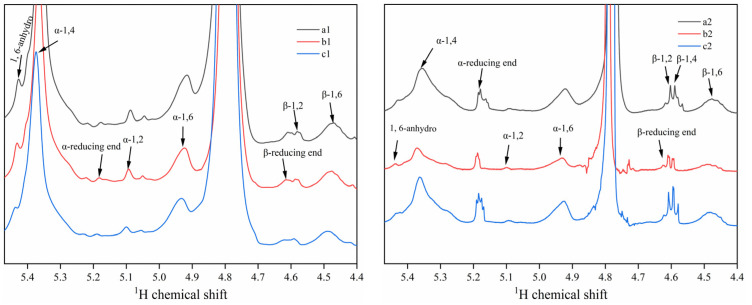
^1^H NMR spectra of resistant dextrins and resistant maltodextrins: a1: sweet potato resistant dextrin; a2: sweet potato resistant maltodextrin; b1: potato resistant dextrin; b2: potato resistant maltodextrin; c1: cassava resistant dextrin; c2: cassava resistant maltodextrin.

**Figure 5 polymers-15-04545-f005:**
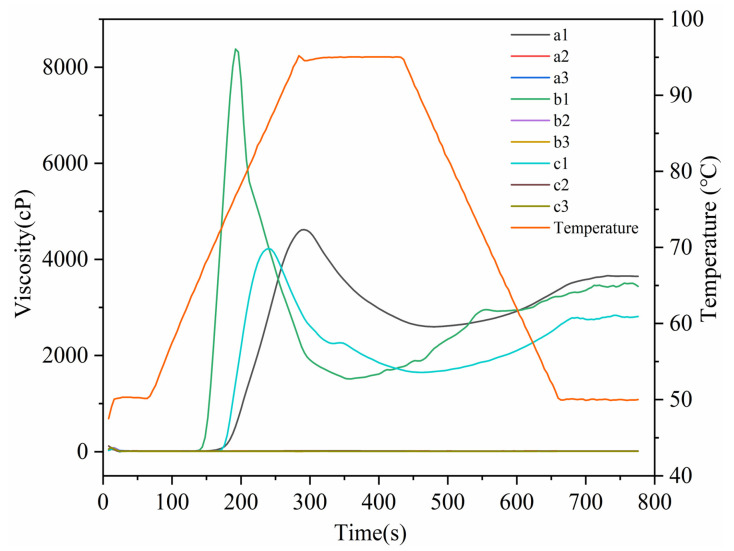
Pasting properties of starches, resistant dextrins, and resistant maltodextrins: a1: sweet potato starch; a2: sweet potato resistant dextrin; a3: sweet potato resistant maltodextrin; b1: potato starch; b2: potato resistant dextrin; b3: potato resistant maltodextrin; c1: cassava starch; c2: cassava resistant dextrin; c3: cassava resistant maltodextrin.

**Figure 6 polymers-15-04545-f006:**
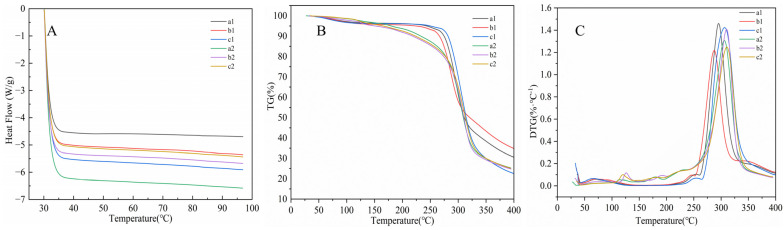
Thermal properties of resistant dextrins and resistant maltodextrins: (**A**) DSC curves, (**B**) TG curves, (**C**) DTG curves. a1: sweet potato resistant dextrin; a2: sweet potato resistant maltodextrin; b1: potato resistant dextrin; b2: potato resistant maltodextrin; c1: cassava resistant dextrin; c2: cassava resistant maltodextrin.

**Figure 7 polymers-15-04545-f007:**
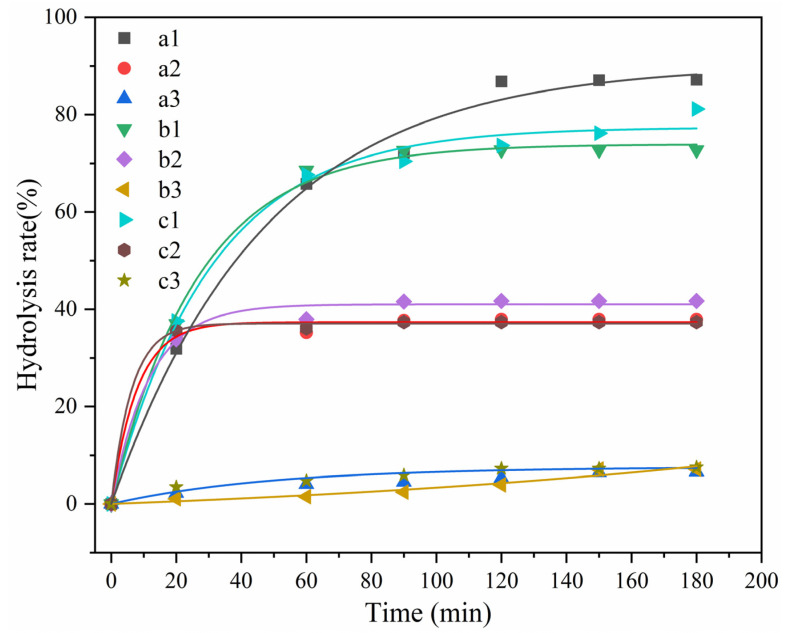
In vitro digestibility of starches, resistant dextrins, and resistant maltodextrins: a1: sweet potato starch; a2: sweet potato resistant dextrin; a3: sweet potato resistant maltodextrin; b1: potato starch; b2: potato resistant dextrin; b3: potato resistant maltodextrin; c1: cassava starch; c2: cassava resistant dextrin; c3: cassava resistant maltodextrin.

**Table 1 polymers-15-04545-t001:** The solubilities and indigestible ingredient contents of resistant dextrins and resistant maltodextrins.

Measured Parameter	Solubility (%)	Indigestible Ingredient Content (%)
a1	64.73 ± 0.35 ^d^	34.32 ± 0.15 ^e^
b1	64.55 ± 0.40 ^d^	36.28 ± 0.98 ^d^
c1	61.27 ± 0.28 ^e^	36.16 ± 1.97 ^d^
a2	97.30 ± 0.17 ^c^	80.60 ± 0.30 ^c^
b2	98.21 ± 0.20 ^b^	84.96 ± 0.52 ^a^
c2	99.82 ± 0.10 ^a^	82.57 ± 0.43 ^b^

a1: sweet potato resistant dextrin; a2: sweet potato resistant maltodextrin; b1: potato resistant dextrin; b2: potato resistant maltodextrin; c1: cassava resistant dextrin; c2: cassava resistant maltodextrin. Different letters indicate significant differences in the same column (*p* < 0.05).

**Table 2 polymers-15-04545-t002:** Characteristic parameters of kinetic equations for in vitro simulated hydrolyses of starch, RDs, and RMDs.

Measured Parameter	C_∞_ (%)	k (10^−2^, min^−1^)	R^2^
a1	90.29 ± 2.45	2.11	0.993
a2	37.35 ± 0.47	12.6	0.994
a3	7.14 ± 0.65	1.32	0.973
b1	73.92 ± 0.92	3.74	0.997
b2	41.03 ± 0.64	8.5	0.992
b3	N/A	N/A	N/A
c1	77.40 ± 1.50	3.25	0.993
c2	37.09 ± 0.23	16	0.998
c3	7.60 ± 0.53	2.02	0.957

a1: sweet potato starch; a2: sweet potato resistant dextrin; a3: sweet potato resistant maltodextrin; b1: potato starch; b2: potato resistant dextrin; b3: potato resistant maltodextrin; c1: cassava starch; c2: cassava resistant dextrin; c3: cassava resistant maltodextrin.

## Data Availability

The data that support the findings of this study are available from the first author, Xinyang Chen, upon reasonable request.

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
