# Peer review of "A Comparative Study of Resistant Dextrins and Resistant Maltodextrins from Different Tuber Crop Starches"

_polymers, 2023, doi:10.3390/polym15234545_

Round 1

Reviewer 1 Report (Previous Reviewer 1)

Comments and Suggestions for Authors

The authors should provide the structural formulas of the studied compounds, the designations of chemical bonds, and schemes of structural modifications.

What do chemical shifts in NMR spectra and vibrational bands in IR spectra refer to?

References: There are no dots in journal abbreviations.

Author Response

Thank you for your valuable suggestions. We have addressed each of them in the document.

Reviewer 2 Report (Previous Reviewer 2)

Comments and Suggestions for Authors

Reviewers' comments have been adequately taken into account.

Author Response

Thank you for your affirmation.

This manuscript is a resubmission of an earlier submission. The following is a list of the peer review reports and author responses from that submission.

Round 1

Reviewer 1 Report

Comments and Suggestions for Authors

The manuscript reports data on the structure and propertiesof three distinct tuber starches and their impact on the anti-digestive characteristics of resistant dextrins and maltodextrins.

The manuscript brings interesting information though the authors should correct several flaws before publishing.

1.      The manuscript completely lacks any structural formulas of the studied compounds, designations of chemical bonds (for example, alpha-1,4, etc.), and schemes of structural modifications, which makes it very difficult to understand the presented results.

2.      Lines 76 and 98: Which reference is correct to describe the method of Bai – 3 or 14?

3.      At the beginning of the Section 3, authors should give a transcript of S1, S2, S3, and then do not repeat this in the captions for each Figure and Table.

4.      Line 191: What does “anomalous protons” mean? Anomeric protons?

5.      Lines 197-199: How exactly do “differences in the preparation methods employed” affect the structure of RDs and RMDs?

6.      Lines 374-375, Conclusion: What is “the theoretical foundation for the applications of RDs and RMDs”?

7.      References are not formatted according to the instructions of the Polymers. There are no DOI in References.

8.      The typos should be corrected, for example, lines 219, 271, 289; the text should be cleaned in general.

The results are worth to be published in Polymers; however, there are several points that the authors have to address before a final recommendation for acceptation can be made.

Reviewer 2 Report

Comments and Suggestions for Authors

1. This are a lot of data in this study but there are major problems, and the manuscript needs important corrections.

2. Line 48: semicolon (;) after “fibers”.

3. The English can often be understood, but needs editing. In particular, there are many errors and inconsistencies in tense, and sometimes the English cannot be understood (see some examples below). Proper language editing and correction is essential.

4. Lines 83, 84: what does “(quality fraction)” mean?

5. Line 85: the sentence reads as if decolorization was a separate step, whereas presumably it occurred during the described process.

6. Section 2.3.5 and line 264: their methods do NOT give the molecular weight distribution, but the molecular SIZE distribution. This is because these are branched molecules; SEC separates by size, not by molecular weight, and branched molecules of the same size but with different branching structures can have different molecular weight. This is a serious error in understanding, which must be corrected. See for example an IUPAC report: “Reliable measurements of the size distributions of starch molecules in solution: current dilemmas and recommendations.” MJ Gidley, I Hanashiro, NM Hani, SE Hill, A Huber, J-L Jane, Q Liu, GA Morris, A Rolland-Sabaté, A Striegel, RG Gilbert. Carbohydrate Polymers 79 255-61 (2010).

7. Section 2.5: it is very common that digestograms follow simple first-order decay, in which case they can be completely represented by a single rate constant. See Butterworth PJ, Warren FJ, Grassby T, Patel H, Ellis PR. Analysis of starch amylolysis using plots for first-order kinetics. Carbohydrate Polymers. 2012;87(3):2189-97; Edwards CH, Warren FJ, Milligan PJ, Butterworth PJ, Ellis PR. A novel method for classifying starch digestion by modelling the amylolysis of plant foods using first-order enzyme kinetic principles. Food & Function. 2014;5(11):2751-8. This must be tested, which is easy with the data presented here.

8. Line 180: “The variations in particle morphology were found to be closely linked to their digestibility”. How was this conclusion reached? It requires quantification of the morphology. This is an important point which must be explained.

9. Line 227: what does “a destination process” mean?

10. Line 121: presumably “Prululan” should be “Pullulan”.

11. Line 296: “the molecular structure of pyrodextrin was dense”: how was this determined?

12. Section 2.3.5: the methods used give very incomplete dissolution of starch, and so the reported molecular weight information is wrong. See the paper by Gidley et al. listed above. Complete molecular dissolution of starch without degradation can only be achieved using dimethyl sulfoxide.

13. Ultrasonication can readily cause molecular degradation, but apparently no check for this was carried out.

Comments on the Quality of English Language

Not good. See above.